# Silver and Gold Complexes with NHC-Ligands Derived from Caffeine: Catalytic and Pharmacological Activity

**DOI:** 10.3390/ijms25052599

**Published:** 2024-02-23

**Authors:** Annaluisa Mariconda, Domenico Iacopetta, Marco Sirignano, Jessica Ceramella, Assunta D’Amato, Maria Marra, Michele Pellegrino, Maria Stefania Sinicropi, Stefano Aquaro, Pasquale Longo

**Affiliations:** 1Department of Science, University of Basilicata, Via dell’Ateneo Lucano 10, 85100 Potenza, Italy; annaluisa.mariconda@unibas.it; 2Department of Pharmacy, Health and Nutritional Sciences, University of Calabria, Via Pietro Bucci, 87036 Arcavacata di Rende, Italy; domenico.iacopetta@unical.it (D.I.); jessica.ceramella@unical.it (J.C.); mariamarra1997@gmail.com (M.M.); michele.pellegrino@unical.it (M.P.); stefano.aquaro@unical.it (S.A.); 3Department of Chemistry and Biology “A. Zambelli”, University of Salerno, Via Giovanni Paolo II 132, 84084 Fisciano, Italy; msirignano@unisa.it (M.S.); asdamato@unisa.it (A.D.); plongo@unisa.it (P.L.)

**Keywords:** xanthine, NHC complexes, silver(I)/gold(I), antitumor, alkyne functionalization

## Abstract

*N*-heterocyclic carbene (NHC) silver(I) and gold(I) complexes have found different applications in various research fields, as in medicinal chemistry for their antiproliferative, anticancer, and antibacterial activity, and in chemistry as innovative and effective catalysts. The possibility of modulating the physicochemical properties, by acting on their ligands and substituents, makes them versatile tools for the development of novel metal-based compounds, mostly as anticancer compounds. As it is known, chemotherapy is commonly adopted for the clinical treatment of different cancers, even though its efficacy is hampered by several factors. Thus, the development of more effective and less toxic drugs is still an urgent need. Herein, we reported the synthesis and characterization of new silver(I) and gold(I) complexes stabilized by caffeine-derived NHC ligands, together with their biological and catalytic activities. Our data highlight the interesting properties of this series as effective catalysts in *A*^3^-coupling and hydroamination reactions and as promising anticancer, anti-inflammatory, and antioxidant agents. The ability of these complexes in regulating different pathological aspects, and often co-promoting causes, of cancer makes them ideal leads to be further structurally functionalized and investigated.

## 1. Introduction

The discovery of the cytotoxic properties of *Cis*platin by Rosenberg [1] has aroused a considerable interest in the development and synthesis of new metal-based drugs for cancer treatment. However, the administration of *Cis*platin, or other platinum-based compounds, is not free of drawbacks that, together with the insurgence of drug resistance, limit its use [2]. These factors highlight the crucial importance of developing new inorganic and organometallic compounds and the extensive study of their biological properties.

During recent decades, numerous inorganic species and organometallic compounds have been synthesized and their antitumoral activity has been evaluated, amongst them ruthenium, iron, palladium, silver, and gold ones [3,4,5,6,7,8]. Based on the obtained results, these compounds, unlike platinum complexes, do not target solely nucleic acids, but their biological actions are also attributable to the inhibition of overexpressed enzymes, present in cancer cells, or the regulation of altered pathways [9,10,11].

In recent years, we have focused our attention on silver and gold compounds [12], which have been designed with ligands that stabilize the oxidation state of gold or ensure a slow release of silver ions, thus guaranteeing a specific action by the metal complexes in the cellular environment.

Recently, these noble metals, which were considered rather chemically inert, were also shown to have significant catalytic activity in the form of complexes stabilized by phosphine or better carbene ligands. Thus, in recent years, silver(I) and gold(I) *N*-heterocyclic carbene (NHC) complexes attracted attention for their biological [13,14] and catalytic activities [15,16,17]. NHCs, also named imidazol-2-ylidene carbenes, can be synthetized by the deprotonation of *N*,*N*′-disubstituted imidazolium salts; moreover, they can be obtained, as well, from caffeine (1,3,7-trimethylxanthine), one of the xanthine derivatives, which presents the methylimidazole moiety in its structure [18]. Xanthine and its derivatives were also studied for their unique structural scaffold and attractive medicinal effects [19,20,21,22] and a silver complex having a caffeine-derived NHC ligand, which has been patented as Silvamist, was proved to possess antibacterial activity against tobramycin-resistant bacteria [23,24,25].

In this paper, we describe the synthesis, characterization, and biological and catalytic activities of new gold(I) and silver(I) complexes stabilized by caffeine derivatives, as NHC ligands. The ligands were obtained by the deprotonation of imidazolium salts, which were characterized by the presence of a hydroxyl group in the substituent of one nitrogen atom and a methyl group on the other. Theophyllines display good solubility in water; furthermore, the presence of a hydroxyl group on one of the nitrogen substituents of the imidazole ring makes the ligand and, consequently, the complex even more soluble. This allows for the use of the compounds as a catalyst even in an ecofriendly solvent, such as water. Furthermore, the interest in the pharmacological field of xanthines can only be enhanced by the formation of the complex that maintains a notable solubility in a physiological environment, also thanks to the hydroxyl present on the substituent of the imidazole ring.

The anticancer activity of the complexes has been evaluated against two breast cancer cell lines, resulting in a higher activity towards the triple-negative MDA-MB-231, where they induced DNA damage, due to the inhibition of the human topoisomerase I (hTopoI), and regulated the NF-κB localization and TNF-α protein levels. Moreover, some of the studied complexes reduced the NO production in RAW 264.7 macrophages, decreasing the iNOS and COX-2 expression levels, with a similar potency of indomethacin, a known molecule with anti-inflammatory activity. Also in this cell context, the newly synthetized complexes were able to reduce the LPS-induced NF-κB nuclear translocation, hampering its transcriptional activity. Some of them showed a remarkable antioxidant activity, as well. Taken together, these outcomes suggest a dual anticancer/anti-inflammatory activity of these Au and Ag complexes, a feature that is important considering the involvement of inflammation and unbalanced redox status in cancer development and progression. Finally, all the tested complexes exerted a discrete antibacterial activity against both Gram+ and Gram− bacterial strains, superior to that of theophylline and caffeine.

Under a chemical point of view, Au(I)/Ag(I) NHC complexes are effective catalysts for some coupling reactions, such as a three-component condensation among an aldehyde, an amine, and a terminal alkyne, named the *A*^3^-coupling reaction, hydroamination, or alkoxylation of alkynes to produce imine or enol, respectively. Thus, the catalytic activity of new synthesized compounds was also evaluated both in *A*^3^-coupling and in hydroamination reactions [26,27,28,29,30,31,32].

These reactions are particularly interesting because they allow for the formation of a C-N bond through environmentally sustainable chemistry: (i) *A*^3^-coupling produces interesting scaffolds such as propargylamines, which are used for the synthesis of various types of molecules (e.g., pyrroles, quinolines, 2-oxazolidinones, etc.) giving water as the only by-product, and (ii) hydroamination of phenylacetylene with anilines produces imines. The latter is another example of a sustainable process, since all the atoms of the reactants are present in the products (imine and enamines) with 100% atom economy.

In the literature, many interesting studies conducted on complexes stabilized by a caffeine-derivate NHC ligand are reported [20,22,33,34], but, to our knowledge, this paper represents the first catalytic application of silver and gold complexes bearing these types of ligands [35].

Overall, the properties of these complexes can be considered intriguing, mostly for the further development of ideal candidates for treating multifactorial and multisymptomatic diseases, such as cancer, or for several applications in chemistry.

## 2. Results and Discussion

### 2.1. Chemistry

*N*-heterocyclic carbene proligands **P-L1**, **P-L2**, and **P-L3** (Figure 1) were obtained following slightly modified synthetic strategies published in the literature [20,22,33,34,36].

Reaction of theophylline with an appropriate alkylating agent, i.e., styrene oxide, cyclohexene oxide, 2-iodoethanol, in the presence of a base (K_2_CO_3_), in dimethylformamide (DMF) at 120 °C, produces the monoalkylated theophylline compounds, which were not isolated and reacted with a large excess of iodomethane (5 eq), giving imidazolium proligands **P-L1**, **P-L2**, and **P-L3**, respectively (Figure 1).

Imidazolium salts were characterized by ^1^H-, ^13^C-NMR, and mass spectroscopy. All the spectra showed the expected signals (see Appendix A). In ^1^H-NMR spectra, the singlet signal attributable to the protons on the carbocationic atom (NC*H*N) resonates at 9.39, 9.54, and 9.37 ppm for **P-L1**, **P-L2**, and **P-L3**, respectively. While, in ^13^C-NMR spectra, the cationic carbons give resonances at 140.93 ppm for **P-L1**, 138.55 ppm for **P-L2**, and 139.63 ppm for **P-L3**. The mass spectra show singlet signals attributable to the carbocationic portion of the xanthinium salts.

Silver acetate complexes **AgL1OAc**, **AgL2OAc**, and **AgL3OAc** were synthesized following the synthetic procedure published by Phillips and Willams et al. by the reaction of imidazolium salt with 2 equivalents of silver acetate at room temperature [20], producing AgI and the desired Ag(NHC)OAc complexes, with yields in the range of 40–55% (Figure 2).

NHC-Silver compounds were characterized by NMR spectroscopy, mass spectrometry, and elemental analysis. In ^1^H-NMR spectra, the absence of the protons around 9.4 ppm confirmed the deprotonation of the cationic carbons of the imidazolium salts and hence the obtainment of the carbene ligands. Furthermore, in such spectra, the appearance of the singlet signal attributable to the methyl group of the acetate anion was evidenced (1.70–1.90 ppm). In the ^13^C-NMR spectra, the resonances of the signals attributable to the carbon bonded to the silver atom (186.36 ppm, 183.85 ppm, and 186.58 ppm for **AgL1OAc**, **AgL2OAc**, and **AgL3OAc**, respectively, Δδ 45–47 ppm) were observed. Finally, the appearance of the signals attributable to the counterion (~175 ppm for *C*=O and ~23 ppm for *C*H_3_C=O) confirmed the formation of the Ag(NHC)OAc complexes. The elemental analysis of all complexes gave the expectable value in C, H, N, and Ag. For all new synthesized silver NHC complexes, the MS analysis showed peaks attributable to a bis-carbene [Ag(NHC)_2_]^+^ structure (at 705.44220, 673.18138, and 583.12378 Da for **AgL1OAc**, **AgL2OAc**, and **AgL3OAc**, respectively). Due to their fluxional behavior, this class of compounds presents, in solution, a dynamic equilibrium between ionic bis-carbenic metal species [(NHC)_2_Ag]^+^[AgX_2_]^−^ and a neutral mono-structure [M(NHC)X] (detailed figures reported in the Appendix A) [37]. Examples of silver complexes having analogous behavior in solution and similar MS analysis are reported in the literature [33,34]. Furthermore, some of us reported an X-ray analysis of an analogous complex, showing a bis-carbene structure [38], while in the literature, solid state structures of neutral mono NHCAgX complexes are described [39], thus supporting the presence of an equilibrium between mono- and bis-carbenic structures.

The gold NHC iodide complexes were synthesized using the procedure developed by Nolan and Gimeno’s research groups [40,41], which prepared gold NHC complexes using a mild and inexpensive base such as K_2_CO_3_ (Figure 3).

The imidazolium salts were reacted with the gold precursor [Au(SMe_2_)Cl] (1.0 eq) in CH_3_CN or acetone, for 3 h, at room temperature. Then, a large excess of the base was added to the solution (10 eq) to obtain the deprotonation of the salt and the coordination of the NHC to the metal center. ^1^H- and ^13^C-NMR spectra show all the expected signals; thus, for silver acetate complexes, the deprotonation was verified through the disappearance of the signal attributable to the proton bonded to the cationic carbon. The coordination of the ligand to the metal center was established by ^13^C-NMR analysis. In fact, the carbene carbons resonate at 184.00, 175.65, and 184.68 ppm for **AuL1**, **AuL2**, and **AuL3**, respectively (Δδ 40–45 ppm).

In order to have a gold complex with the same counterion as the silver ones, and thus have a more homogeneous comparison between the catalytic and biological activities of two complexes that have the same ligands and differ only in the metal center, the gold NHC acetate complex bearing ligand L1 was synthesized. This was obtained by the reaction of the gold NHC iodide complex with silver acetate in dry dichloromethane, following the procedure published by Tacke et al. [42] and modified by us [12] (Figure 4).

The NHC gold acetate was characterized by ^1^H and ^13^C-NMR spectroscopy, mass spectrometry, and elemental analysis. The ^1^H-NMR spectrum shows all the expected signals; the singlet signals attributable to the protons of the methyl group of the acetate anions (*δ* 1.79 ppm in DMSO-d_6_) confirm the formation of the gold acetate compound. Further confirmation is provided by the ^13^C-NMR spectrum which shows the signals of two carbon atoms of the anions (*δ* 173.90 ppm for CH_3_***C***=O and *δ* 22.81 for ***C***H_3_C=O) and the upfield shift of the signal attributable to the carbene carbon atom, i.e., at 184.00 ppm for **AuL1** and 168.38 for **AuL1OAc**.

Congeners **AuL1** and **AuL1OAc** complexes were also characterized via MS analysis. As seen for silver complexes, and reported in the literature for analogous gold complexes [12,43,44], the MS spectra showed peaks attributable to [(NHC)_2_Au]^+^ structures at 825.27616 Da for both complexes. The MS analyses for **AuL2** and **AuL3** gave peaks at 781.27614 Da and 673.18217 Da, respectively (detailed figures reported in the Appendix A).

Hydrolysis tests provide crucial information on the stability of the complexes; thus, to evaluate the hydrolytic stability of the synthesized complexes, two of them, which were then found to be among the most active in catalysis and among the most pharmacologically interesting, were subjected to hydrolysis tests, i.e., **AuL1** and **AgL2OAc**. Since we can expect that the rapid hydrolysis of the leaving group (-I or -OAc) and NHC ligands could give biologically inactive species, instead, an active species could be generated if the NHC ligands remain metal bound. Stability tests were conducted on complexes **AuL1** and **AgL2OAc** to verify the stability of hydrolysis in the presence of chloride ions, i.e., 5 mg of complex was solubilized in 0.5 mL of DMSO-d6 and 0.05 mL of a saturated solution in D2O of NaCl was added to the solution. The spectrum of the complex does not change even after 90 h.

The data obtained provide sufficient evidence that the presence of NHC groups is effective for the stabilization of the complexes. Therefore, these coordinating groups may be critical to their biological efficacy.

#### Catalytic Activity in A^3^-Coupling and Hydroamination Reactions

Silver and gold complexes bearing *N*-heterocyclic carbene ligands, due to their carbophilicity [45], are able to catalyze functionalizations such as the hydration [46,47], alcoxydation [48,49], hydroamination [50,51,52], and hydroarylation [53] of terminal and internal alkynes; the carboxylation of terminal alkynes [54,55]; and other interesting reactions [17,45,56].

In recent years, we have synthesized a series of silver and gold(I) complexes bearing NHC ligands and evaluated their catalytic activity in *A*^3^-coupling and hydroamination reactions, with detailed insight in their catalytical mechanism [32,43,44]. This type of reaction is a condensation among an aldehyde, an amine, and a terminal alkyne to gain propargylamines [27,28,57]. The latter is an important and versatile class of precursor molecules in the synthesis of various organic compounds such as pyrroles, pyrrolidines, imidazoles, etc. [26,27]. Furthermore, propargylamines such as pargyline, rasagiline, and selegiline are used for the treatment of the first stages of neurodegenerative disorders, i.e., Alzheimer’s and Parkinson’s diseases.

Thus, we tested the catalytic activity of silver and gold(I) complexes in *A*^3^-coupling reactions using three different aldehydes (*p*-formaldehyde, cyclohexylcarbaldehyde, and benzaldehyde) with piperidine and phenylacetylene, to lead the corresponding propargylamines (see Figure 5).

To the best of our knowledge, this is the first example reported in the literature of the catalytic activity promoted by caffeine derivative NHC silver and gold(I) complexes. Table 1 reports the yields of the obtained propargylamines, evaluated by ^1^H-NMR, using 2-bromomesitylene as an internal standard. This assessment is performed by integration of the standard singlet signal (at 6.89 ppm, s, 2H) vs. the signal of the propargylic proton(s) [at *δ* 3.43, s, 2H for *N*-(3-phenyl-2-propynyl)piperidine; *δ* 3.11, s, 1H for *N*-(1-cyclohexyl-3-phenyl-2-propynyl)piperidine; and *δ* 4.79, s, 1H for *N*-(1,3-diphenyl-2-propynyl)piperidine].

From an analysis of the data reported in the Table 1, it may be useful to underline the following:All the reactions were carried out without the use of solvents, so they are also interesting from the point of view of environmental protection;All the complexes are active in this type of catalytic transformation. NHC-gold(I) iodide complexes have shown better catalytic activity than the silver acetate analogue, for all three types of aldehydes;**AuL1OAc** is more active than silver analogues, as can be observed by comparing the results of runs 1–3 vs. 13–15;*p*-formaldehyde is almost completely converted in the product, except in run 7, where the complex **AgL3OAc** was used as a catalyst;Aliphatic aldehydes are more reactive than aromatic aldehydes;**AuL1OAc** is less reactive than the iodide analogues, as can be observed by comparing the results of runs 10–12 vs. 12–15.

Caffeine-derived NHC gold complexes were also tested in the hydroamination reaction of phenylacetylene with anilines, to produce the corresponding imine (Figure 6).

The hydroamination reaction is a coupling between an amine with an alkyne or alkene to gain a C-N bond with 100% atom economy, being all the atoms of reactants present in the products [30,31,58]. The reaction is co-catalyzed, by a silver salt, in order to remove the anion coordinated to the metal center by precipitation. The catalytic activity of the gold complex is strongly influenced by the choice of the silver salt. The best reactivity was found using silver salts with non-coordinating anions [32,51]; thus, silver hexafluoroantimonate (AgSbF_6_) was chosen as the halogen scavenger. The reactions were conducted in dry acetonitrile, for 16 h at 90 °C, under a nitrogen atmosphere. The results are listed in Table 2. The yields of imines were determined by ^1^H-NMR, using the internal standard method, by the integration of the methyl protons of the product at 2.24 ppm with the singlet signal of the standard (dibromomethane) at δ 4.93 ppm in CDCl_3_.

Observing the data in Table 2, it is possible to establish that (i) all the examined gold NHC complexes are able to catalyze the addition of aniline to the phenylacetylene; (ii) the counterion of the NHC-Au is not crucial in determining the activity of catalysts (cfr. run 1 with run 4); (iii) as previously reported by some of us [32], on the NHC ligand the (2-hydroxy-2-phenyl)ethyl substituent of one of the nitrogen atoms is the most effective for catalytic activity; and, lastly, (iv) *para*-aniline substitution is detrimental in terms of substrate reactivity.

### 2.2. Anticancer Activity

The anticancer activity of the new synthesized complexes was evaluated using the MTT cell viability test on two different breast cancer cell lines: the ER+ MCF-7 and the triple-negative MDA MB-231. The obtained results, listed in Table 3, indicate that the highest anticancer activity was recorded under the exposure of **AgL1OAc**, **AuL1**, and **AgL2OAc** against both the breast cancer cells. In particular, these complexes resulted as more active on the highly aggressive and metastatic MDA-MB-231 line, where **AuL1**, **AgL2OAc**, and **AgL1OAc** showed IC_50_ values of 14.7 ± 1.1, 19.4 ± 1.0, and 32.3 ± 0.9 µM, respectively. The same complexes resulted as less active on the MCF-7 cells; indeed, **AgL2OAc** showed an IC_50_ value 28.7 ± 1.0 µM, followed by **AgL1OAc** and **AuL1** with IC_50_ values of 39.6 ± 0.8 and 73.8 ± 1.1 µM, respectively. The anticancer activity of caffeine and theophylline, two xanthine derivatives used as reference compounds, has also been evaluated and, as visible in Table 3, they did not show anticancer activity on both breast cancer lines up to a concentration of 500 µM and under the experimental conditions used in our assay.

The selectivity of the studied complexes was evaluated, as well, by determining the possible toxicity towards the human mammary epithelial cell line MCF-10A. Table 3 clearly shows that the exposure to the complexes did not interfere with the normal cell growth, at least up to a concentration of 100 µM, as well as for caffeine and theophylline (>500 µM). On the contrary, the platinum-based chemotherapy agent *Cis*platin, used also as a reference compound, showed a lower activity than **AuL1** and **AgL2OAc** or comparable to **AgL1OAc** on the MDA-MB-231 (IC_50_ = 32.2 ± 1.0 µM) and, at the same time, a higher cytotoxicity on the normal MCF-10A cells (IC_50_ = 78.2 ± 1.2 µM). Finally, proligands **P-L1-3** were also tested on all the cell lines used, showing no activity until the concentration of 100 µM (Appendix A).

In order to evaluate whether the observed anticancer activity of **AuL1** and **AgL2OAc**, the most active complexes, against MDA-MB-231 was related to their ability to induce DNA damage, we performed the TUNEL assay. The MDA-MB-231 cells were exposed to the tested complexes used at a concentration of 30 µM for 24 h and, subsequently, further processed and imaged as described in Section 3. The obtained results are shown in Figure 2. In the cells treated with **AuL1** and **AgL2OAc**, it was possible to observe the formation of a green nuclear fluorescence due to the action of the rTdT enzyme that catalyzed the incorporation of the fluorescein-12-dUTP at the 3′-OH DNA fragments formed as a result of the DNA damage (Figure 2, panel B, **AuL1** and **AgL2OAc**). On the contrary, in the breast cancer cells treated with only the vehicle, the absence of the green nuclear fluorescence confirmed that no DNA fragmentation occurred (Figure 2, panel B, CTRL). Panel A represents the DAPI for cell nucleus staining. Aiming at further understanding the mechanism underlying the observed DNA damage induced by the lead complexes, we performed a human topoisomerase I (hTopoI) inhibition assay. hTopoI is mainly involved in controlling the DNA topology during its different metabolic phases [12] and is strictly needed for sustaining the uncontrolled and rapid cell division, a phenomenon occurring during cancer onset and progression. Since the inhibition of hTopoI can provoke DNA double-strand breaks and cell death [59], the discovery of new inhibitors is very desired in the fight against cancer. To assess the ability of the lead complexes **AuL1** and **AgL2OAc** in inhibiting hTopoI, we performed an in vitro relaxation assay. In this assay, we exposed hTopo I to each lead, in the presence of a supercoiled DNA (SC DNA) as a substrate, and analyzed the reaction products by the means of agarose gel electrophoresis. The outcomes (Figure 3) indicated that both the complexes totally inhibited the hTopo I supercoil relaxing activity at the concentration of 1 μM. Indeed, the uncut SC DNA band is present at the bottom of the gel (Figure 3, lanes **AuL1** and **AgL2OAc**). Of course, hTopo I retained full activity in the presence of only the vehicle (DMSO, Figure 3, lane CTRL), cutting the SC DNA and producing the relaxed DNA topoisomers. As reference molecules, *Cis*platin and caffeine were used; however, only the first inhibited the hTopoI activity, but at the concentration of 50 μM, whereas caffeine failed at 1 μM (lanes CP and C, respectively, Figure 3). Relaxed and supercoiled pHOT1 DNAs were used as markers (Figure 3, M and SC DNA, respectively).

### 2.3. Anti-Inflammatory Activity

The anti-inflammatory activity of the new synthesized complexes was evaluated by performing the Griess assay, using the RAW 264.7 murine macrophages. It is well known that following an inflammatory stimulus, macrophages produce nitric oxide (NO) and other pro-inflammatory cytokines (TNF-α, interleukins, and so on). In particular, NO plays a fundamental role as a second messenger and is produced by the nitric oxide synthase (NOS) enzymes [60]. Using the Griess assay, we investigated the ability of all the complexes to interfere with the NO production. We used caffeine and theophylline as reference compounds and indomethacin (I), a nonsteroidal anti-inflammatory drug (NSAID), as a positive control. Furthermore, the activity of **P-L1-3** proligands was determined and the results were reported in the Appendix A. Lipopolysaccharide (LPS), at a concentration of 1 µg/mL, was used as a NO production inducer in RAW 264.7 cells. Viability assays (MTT) were preliminarily performed (25 µM, 24 h) in order to exclude any cytotoxic effect of the tested complexes against the RAW 264.7 cells (97 to 95% of residual viability, treated vs. CTRL, Appendix A).

Figure 4 showed that the RAW 264.7-LPS treated cells produced NO and that indomethacin, at a concentration of 25 µM, reduced the NO levels by about 11.23%. At the same concentration, the complexes **AgL3OAc**, **AuL2**, and **AuL3** showed a similar ability to that of indomethacin in reducing the NO production, being about 14.35, 12.64, and 14.12%, respectively. The complexes **AgL2OAc** and **AuL1** also produced a reduction in NO production of about 5.20 and 6.23%, but it should also be evidenced that they were the most active in diminishing the breast cancer cells’ viability. Thus, these results appoint the **AuL3**, **AgL3OAc**, and **AgL2OAc** complexes as the most active anti-inflammatory agents, with an activity similar to indomethacin. At the same time, the dual anticancer and anti-inflammatory activity exerted by **AgL2OAc** and **AuL1** make them potential candidates for the development of beneficial synergistic drugs. Indeed, it must be recalled that inflammatory responses may be elicited by anticancer therapies and that the inflammation is closely associated with cancer progression, providing an optimal microenvironment for metastasis development [61]. Finally, **P-L1-3** proligands were found to be inactive in reducing NO production (Appendix A).

A key mediator of inflammation is represented by the inducible nitric oxide synthase (iNOS), which produces nitric oxide (NO) from L-arginine and is crucial for the physiological inflammatory reaction. Moreover, the overexpression and/or dysregulation of iNOS have been involved in different pathologies, including cancer, neurodegeneration, and so on [62]. Therefore, using immunofluorescence studies, we evaluated whether the complexes **AuL3** and **AgL3OAc**, the most able to reduce LPS-induced NO levels, may also be able to regulate iNOS expression in RAW 264.7 macrophages. For this purpose, the latter were exposed to the vehicle alone (DMSO) or LPS, or they were co-treated with LPS (1 µg/mL) and indomethacin (positive control) or **AuL3** or **AgL3OAc**, used at a concentration of 25 µM. The results, reported in Figure 5, evidence that the LPS-stimulated macrophages possess higher iNOS expression, as indicated by the presence of a marked red fluorescence (Figure 5a, panel 1B, and Figure 5b, where the fluorescence quantification is graphed). Contrarily, the cells treated with the vehicle alone, lack of this response (Figure 5a, panel 2B). In the cells exposed simultaneously to LPS and indomethacin, the iNOS levels were lower, reaching those obtained with the CTRL cells (Figure 5a, panel 3B, and Figure 5b). In a comparable manner, **AgL3OAc** was able to reduce the iNOS expression levels with a similar potency of indomethacin, as confirmed by a net reduction in iNOS expression (Figure 5a, panel 4B and Figure 5b). **AuL3** produced a similar effect, although to a lesser extent than that of indomethacin and **AgL3OAc** (Figure 5a, panel 5B, and Figure 5b).

Together with iNOS, COX-1 and COX-2 are important pro-inflammatory proteins involved in the synthesis of prostaglandins (PGs) from arachidonic acid and are inhibited by nonsteroidal anti-inflammatory drugs (NSAIDs) [63]. It is well known that COX-1 is constitutively expressed in most human tissues, while COX-2 expression is induced by inflammatory stimuli, including LPS and cytokines, in macrophages [64,65].

To evaluate the regulatory effect of **AgL3OAc** and **AuL3** on COX-1 and COX-2 expression in the LPS-stimulated RAW 264.7 macrophages, immunofluorescence studies were used, under the same experimental condition adopted for iNOS experiments. Thus, RAW 264.7 cells were treated with DMSO (CTRL), with LPS (LPS, 1 µg/mL), or co-treated with LPS and indomethacin (positive control), **AgL3OAc**, or **AuL3**, at 25 µM.

The obtained results (Figure 6) clearly show that the treatment of RAW 264.7 macrophages with LPS did not induce any significant increase in COX-1 protein expression (Figure 6a, panel 2B). Similarly, the co-treatment with LPS and indomethacin, **AgL3OAc**, or **AuL3** (Figure 6a, panels 3, 4, and B, respectively) produced no difference in protein expression level when compared to cells treated with the vehicle alone (Figure 6a, panel 1B). Figure 6b shows the quantification of the fluorescence related to COX-1 expression. These data are in agreement with the literature regarding COX-1, since it is a constitutive isoform, is expressed in most tissues under basal conditions, and its expression is not induced by LPS [66].

Contrarily, the treatment with LPS (1 μg/mL) remarkably increased COX-2 protein expression in RAW 264.7 macrophages (Figure 7a, panel 2B) if compared with the vehicle-treated cells (Figure 7a, panel 1B), as indicated by the green fluorescence associated with COX-2. As expected, the green fluorescence was, instead, reduced by about 30% in the presence of indomethacin (25 μM) (Figure 7a, panel 3B). **AgL3OAc** showed a superior anti-inflammatory activity compared to indomethacin at the same concentration, inducing a remarkable reduction in COX-2 expression of about 60% (Figure 7a, panel 4B). Even if to a lesser extent, **AuL3** also decreased COX-2 expression, by about 40%, but it resulted as more active than indomethacin at the same concentration (Figure 7a, panel 5B). The COX-2-associated green fluorescence was quantified using ImageJ software 1.54d and the results are reported in Figure 6b. As it is known, COX-2 is an inducible enzyme that produces prostaglandins and is usually considered responsible for inflammation in pathological states [67]. Moreover, COX-2 has been found highly expressed in invasive estrogen-independent breast cancer cells and is believed to be involved in cancer progression [68]. Thus, the development of compounds that hamper the selective expression of COX-2 is highly desired for slowing down cancer progression.

In the last few decades, the contribution of inflammation to cancer development and progression has gained great success, with a particular focus on some mediators strictly involved in both diseases [69]. Among them, NF-κB activation represents an important hallmark of cancer and inflammation due to its ability to regulate the expression of numerous genes implicated in cell proliferation, angiogenesis, apoptosis, inflammation, invasion, and metastasis [70]. Thus, modulation of the NF-κB signaling pathway could be useful in the treatment of both cancer and inflammatory pathologies.

Normally, in most types of cells, NF-κB, combined with IκB, is predominantly cytoplasmic, but when cells are subjected to various stimuli, IκB is degraded and NF-κB migrates into the nucleus where it becomes transcriptionally active [70]. Aiming at the evaluation of the status of NF-κB in both the cell models previously described, we again adopted immunostaining studies. Particularly, NF-κB intracellular localization was studied in MDA-MB-231 breast cancer cells treated with **AuL1** or **AgL2OAc** and also in the RAW 264.7 macrophages treated with **AgL3OAc** or **AuL3**.

We exposed the MDA-MB-231 cells to the vehicle alone (DMSO), **AuL1**, or **AgL2OAc**, at the concentration of 15 µM. As visible in the MDA-MB-231 cells treated with **AuL1** or **AgL2OAc**, the NF-κB (red fluorescence) is prevalently located in the cytoplasm, as indicated by the white arrows (Figure 8, panel B, **AuL1** and **AgL2OAc**, respectively), whereas the vehicle-treated cells exhibited a prevalent nuclear localization (Figure 8, panel B, **CTRL**, see white arrows), where it regulates the gene expression needed for cancer progression. Thus, both the complexes were able to cause the inhibition of NF-κB translocation from the cytoplasm to the nucleus of MDA-MB-231 cells, hampering its transcriptional activity. Several studies showed that the endogenous activity of NF-κB is often higher in breast cancer and clearly associated with the induction of target genes that can inhibit apoptosis [69]. Thus, this evidence supports the observed apoptosis triggered by the complexes **AuL1** and **AgL2OAc** in MDA-MB-231 cells.

For a further confirmation, the NF-κB status was also checked in RAW 264.7 cells. Thus, the latter were treated with the vehicle alone (DMSO, CTRL), with LPS alone (LPS), co-treated with LPS and indomethacin (positive control), or with LPS and **AuL3** or **AgL3OAc** at 25 µM. Figure 9 suggests that in the RAW 264.7 cells treated with the vehicle alone, NF-κB was mainly localized in the cell cytoplasm (Figure 9, panel 1B). Conversely, the treatment with LPS strongly induced the translocation of NF-κB from the cytosol to the nucleus (as indicated by the white arrows) where the red fluorescence appeared more intense (Figure 9, panel 2B). The macrophages simultaneously exposed to LPS and indomethacin exhibited, instead, a higher cytoplasmic localization of NF-κB, confirming the anti-inflammatory properties of the NSAID, used as a positive control (Figure 9, panel 3B, see white arrows). As expected, both **AgL3OAc** and **AuL3** showed a similar behavior of indomethacin, since NF-κB localization is mostly cytoplasmic, as evidenced by the white arrows, meaning that both complexes were able to reduce the LPS-induced NF-κB nuclear translocation (Figure 9, panels 4 and 5B, respectively). Taken together, the presented results about the NF-κB modulation are reproducible in the two employed cellular contexts, corroborating the stated dual anticancer and anti-inflammatory activity of these complexes.

Finally, given that the NF-κB pathway is strictly related to the proinflammatory tumor necrosis factor-α (TNF-α) in promoting cancer cells’ proliferation, the inhibition of apoptosis, angiogenesis, and metastasis, creating a complex link between inflammation and cancer [71], we determined the TNF-α expression in MDA-MB-231 cells in the presence, or not, of the studied complexes.

Even if the TNF-α transduction pathway is complex, not completely elucidated, and still debated, the binding to its receptor induces different events, amongst them the IkB phosphorylation, ubiquitination, and degradation, followed by the release of the transcription factor NF-κB and its translocation to the nucleus [70].

The MDA-MB-231 cells were exposed for 24 h to the vehicle alone (DMSO, CTRL) or **AuL1** or **AgL2OAc** (15 µM). TNF-α immunostaining suggests a diminished expression in the cells treated with the complexes, mostly for **AuL1**, if compared with the cells exposed to DMSO (Figure 10a, panel B). Indeed, the fluorescence quantification (Figure 10b) confirmed that the cells treated with **AuL1** or **AgL2OAc** revealed reductions of about 60 and 50% in TNF-α expression, respectively. This is another important effect elicited by the complexes under investigation, since TNF-α can act as an endogenous cancer promoter, bridging inflammation and carcinogenesis, and has been found highly expressed in different pre-neoplastic and tumor tissues [72].

### 2.4. Antioxidant Activity

Xanthine derivatives are known to possess interesting antioxidant and radical scavenging activities that contribute to their biological properties, including anticancer and anti-inflammatory activities. In order to evaluate the in vitro capability of the new synthesized complexes to inhibit reactive oxygen species (ROS) production, a cell-based assay employing the dihydro-2′,7′-dichlorofluorescein diacetate (H_2_DCF-DA) fluorescent probe was performed. In particular, the murine fibroblasts BALB/3T3 were pre-treated for 24 h with all the complexes used at a concentration of 25 µM, at which no toxicity was recorded, and subsequently, the ROS production was stimulated using menadione. N-acetyl-cysteine (NAC), a well-known and powerful antioxidant compound, was used as a positive control at a concentration of 20 mM.

The obtained results, reported in Figure 11, showed that the best antioxidant activity was recorded under the exposure of **AuL1**, as it produced an ROS level reduction of approximately 69%, greater than that recorded following the treatment with NAC (approximately 63%). However, it is important to underline that all the complexes were tested at the concentration of 25 µM, which is much lower than that of NAC. **AgL1OAc**, **AgL2OAc**, and **AuL1OAc** showed a remarkable antioxidant activity as well, producing reductions of about 61, 59, and 54%, respectively, of ROS levels. Moreover, all the synthesized complexes exerted an antioxidant capacity higher than caffeine and theophylline, used as reference compounds.

In addition, in order to study the antioxidant properties of these complexes in depth, two spectrophotometric assays were carried out for evaluating the scavenging activity of the complexes towards two stable radicals, such as ABTS and DPPH. Trolox, a water-soluble analogue of vitamin E, well known for its antioxidant properties, was used as a control, and caffeine and theophylline were used as reference compounds.

The calculated IC_50_ values, expressed in µM, are reported in Table 4. First, none of the tested complexes was able to act as a scavenger of the DPPH radical (IC_50_ > 500 uM). On the contrary, Trolox exhibited excellent antioxidant properties, with an IC_50_ value of 47.03 ± 1.2 µM.

As regards the ABTS assay, the obtained results demonstrated that **AgL3OAc**, **AuL1OAc**, and **AuL3** were not able to act as scavengers of the ABTS radical, at least up to a concentration of 500 µM. However, the other complexes appear to have a certain scavenging activity towards ABTS, although lower than that of Trolox. Indeed, the recorded IC_50_ values were 145.4 ± 1.1, 149.2 ± 0.7, 155.5 ± 1.0, and 161.0 ± 1.0 µM for **AgL2OAc**, **AgL1OAc**, **AuL2**, and **AuL1**, respectively. Again, Trolox showed an excellent ability to scavenge the ABTS radical, with an IC_50_ value of 54.49 ± 0.9 µM. Finally, caffeine and theophylline did not show any scavenging capacity towards either of the radicals used in these assays (Table 4). Overall, **AuL1** was found to be the best ROS-scavenging complex in the considered cellular context, followed by **AgL1OAc** and **AgL2OAc**, whereas the ability in scavenging the ABTS radical was very similar and null for DPPH.

### 2.5. Antibacterial Activity

Finally, considering some data that report the ability of caffeine and some derivatives to inhibit many strains of pathogenic bacteria [73], the new synthesized complexes were investigated as potential antibacterials as well.

Their activity was tested on different bacterial strains, both Gram-negative (*Escherichia coli*, *Klebsiella pneumoniae*, *Pseudomonas aeruginosa*, and *Salmonella typhimurium*) and Gram-positive (*Staphylococcus aureus*, *Enterococcus faecalis*, and *Staphylococcus epidermidis*). The obtained results, in terms of the minimum inhibitory concentration (MIC) and minimum bactericidal concentration (MBC), expressed as µg/mL, are listed in Table 5. All the tested complexes exerted a discrete antibacterial activity; however, all were superior to that of theophylline and caffeine, used as reference compounds, that demonstrated a total inactivity up to a concentration of 200 µg/mL. Moreover, the used strains were sensitive to Ampicillin [74,75], Gentamicin [76,77,78], and Cephalosporin [79,80] (see Table 5), accordingly to the literature data. In particular, all of them exhibited a similar activity on *E. coli* and *S. aureus*, where the Ag complexes resulted as more active: **AgL2OAc** was found to be the most active, with an MIC of 100 µg/mL, followed by **AgL1OAc** and **AgL3OAc** (MIC = 125 µg/mL) and **AuL1** and **AuL2** (MIC = 150 µg/mL). An interesting antibacterial activity for both Ag and Au complexes bearing the ligand L2 was recorded on *P. aeruginosa* and *S. epidermidis*, where they resulted as more active, with an MIC of 100 µg/mL. The same MIC value was found for **AgL3OAc** on *S. epidermidis* and **AuL1** on *P. aeruginosa*. In addition, **AuL1** and **AgL1OAc** showed a slightly lower antibacterial activity against *S. epidermidis* (MIC 125 µg/mL), as well as **AgL1OAc** and **AgL3OAc** against *P. aeruginosa* (MIC 125 and 150 µg/mL, respectively). A comparable antibacterial activity was also found on *K. pneumoniae*, where **AgL1OAc** and **AgL2OAc** exhibited an MIC of 125 µg/mL, while **AuL1**, **AuL2**, and **AgL3OAc** produced a higher MIC value, equal to 150 µg/mL. Instead, a lower antibacterial activity was recorded on *S. typhimurium* and *E. faecalis*. Finally, **AuL1OAc** and **AuL3** complexes resulted to be inactive on all the bacterial strains used in this assay, at least up to a concentration of 200 μg/mL and under the adopted experimental conditions. Overall, the new complexes exhibited a moderate activity if compared with the most potent and clinically used reference compounds, for each studied strain; however, the frequent bacterial resistance phenomena onset and the other biological properties previously described make them interesting compounds to be further developed.

## 3. Materials and Methods

### 3.1. Chemistry

All reactants were purchased by TCI chemicals (Zwijndrecht, Belgium) and used as received. The solvents (Carlo Erba Reagents srl, Milano, Italy) were dried under a nitrogen atmosphere by heating at boiling temperature over suitable drying agents. Deuterated solvents were degassed under a nitrogen atmosphere and stored in the dark, over activated 4 Å sieves. NMR characterizations were carried out at 298 K on a Bruker AVANCE 400 (Milano, Italy) spectrometer (400 MHz for ^1^H; 100 MHz for ^13^C) and a Bruker AM 300 (Milano, Italy) spectrometer (300 MHz for ^1^H; 75 MHz for ^13^C). The chemical shifts refer to tetramethylsilane (SiMe_4_, *δ* = 0) using the residual proton impurities of the deuterated solvents (Eurisotop Cambridge Isotope Laboratories, Cambridge, UK) as internal standards (*δ* 2.50 for DMSO, *δ* 5.32 for CD_2_Cl_2_, *δ* 7.26 for CDCl_3_), and in ^13^C NMR spectra, they use the following solvent peaks at *δ* 39.51 for DMSO, *δ* 54.00 for CD_2_Cl_2,_ and *δ* 77.23 for CDCl_3_. The multiplicities are abbreviated in the following manner: singlet (s), doublet (d), triplet (t), multiplet (m), broad (br), and overlapped (o). Elemental analyses for C, N, and H were obtained according to standard microanalytic procedures and recorded using a Thermo-Finnigan Flash EA 1112. The quantities of halogens (Cl, I) were calculated by the reaction of AgNO_3_ with the halogen and the precipitation of the silver halide (AgX). Then, it was dissolved in Na_2_S_2_O_3,_ and the silver content was determined by atomic flame absorption spectroscopy (FAAS), achieving the halogen content from the amount of silver.

ESI-MS spectra were recorded using a Waters Quattro Micro triple-quadrupole mass spectrometer equipped with an electrospray ionization source. MALDI-MS spectra were obtained using a Bruker SolariX XRF Fourier transform ion cyclotron resonance mass spectrometer (Bruker Daltonik GmbH, Bremen, Germany) equipped with a 7 T refrigerated actively shielded superconducting magnet (Bruker Biospin, Wissembourg, France). The mass range was set to *m*/*z* 200–3000. The laser power was 28% and 22 laser shots were used for each scan. The mass spectra were calibrated externally using a mix of peptide clusters in the MALDI positive ionization mode. The accuracy of the mass was improved by the internal recalibration by matrix ionization (2,5-dihydroxybenzoic acid).

#### 3.1.1. General Procedure for the Synthesis of Xanthinium Salts (P-L1, P-L2, and P-L3)

Xanthine-derived imidazolium salts were synthesized following the procedure published in the literature [22,34,35,81] and applying the synthetic strategies developed by some of us [36,38,43,44]. Theophylline (1.0 eq) was reacted with potassium carbonate (K_2_CO_3_, 2.0 eq), in DMF (40 mL) for 2 h at 120 °C, for the deprotonation of the sp^3^ hybridized nitrogen atom. To the suspension was added 1.2 eq of appropriate alkylating agent (styrene oxide for P-L1, cyclohexeneoxide for P-L2, and 2-iodoethanol for P-L3). The mixtures were stirred for 24 h at 120 °C, for the alkylation of the first deprotonated nitrogen atom. Subsequently, the reaction mixtures were filtered to remove the excess of the base and iodomethane (5 eq) was added to the solutions. The resulting mixtures were stirred for 6 h at room temperature. The xanthinium salts were obtained by removing the solvent in vacuo and washing the residue with hexane (3 × 100 mL).

##### Synthesis of P-L1, 9-[(2-Hydroxy-2-phenyl)ethyl]-1,3,7-trimethylxanthinium Iodide

Theophylline (2.00 g, 11.0 mmol) was dissolved in DMF (40 mL) in a round-bottom flask under an inert atmosphere. K_2_CO_3_ (3.04 g, 22.0 mmol) was added to the solution which was stirred for 2 h at 120 °C. Then, phenylethylene oxide (1.32 g, 13.2 mmol) was added. The resulting mixture was stirred for 24 h, filtered to eliminate the excess of the base, then iodomethane (7.75 g, 55.0 mmol) was introduced. The solution was kept under stirring for 6 h. The imidazolium salt (3.53 g, 8.00 mmol) was obtained after the removal of the solvent under reduced pressure and washing with hexane (3 × 100 mL).

Yield: 72%.

^1^H-NMR (400 MHz, DMSO-d_6_): *δ* 9.39 (s, 1H, NC*H*N), 7.44–7.30 (m, 5H, *Ph-group*), 5.96 (s, 1H, O*H*), 4.73 (s, 1H, C*H*OH), 4.69–4.40 (m, 2H, NC*H*_2_), 4.35 (s, 3H, NC*H*_3_ imidazolium ring), 3.74 (s, 3H, NC*H*_3_), 3.27 (s, 3H, NC*H*_3_).

^13^C-NMR (100 MHz, DMSO-d_6_): *δ* 153.33 and 150.19 (*C*=O *purine ring*), 140.93 (*backbone carbon*, CH_3_N*C*=C), 139.93 (N*C*HN), 139.55 *(ipso aromatic carbon*, Ph-ring), 128.26 (*aromatic carbons*, Ph ring), 128.21 (*aromatic carbon*, Ph ring), 125.87 (*aromatic carbons*, Ph ring), 108.82 (*backbone carbon*, C=*C*NCH_2_), 70.30 (*C*HOH), 55.45 (N*C*H_2_), 36.93 (N*C*H_3_ imidazolium *ring*) 31.48 and 28.36 (N*C*H_3_ purine ring).

ESI-MS (CH_3_CN *m*/*z*) = 315.14594 attributable to the cationic part of imidazolium proligand [C_16_H_19_N_4_O_3_]^+^.

Elemental Analysis: calculated C 43.45, H 4.33, I 28.69, N 12.67, O 10.85; experimental C 43.40, H 4.30, I 28.68, N 12.80, O 10.81.

##### Synthesis of P-L2, 9-[Cyclohexan-2-ol]-1,3,7-trimethylxantinium Iodide

Under a nitrogen atmosphere, theophylline (2.00 g, 11.0 mmol) and K_2_CO_3_ (3.04 g, 22.0 mmol) were suspended in DMF (40 mL) and stirred for 2 h. Cyclohexene oxide (1.30 g, 13.2 mmol) was added to the reaction mixture, which was kept under stirring for 24 h at 120 °C. Afterwards, it was filtered to remove the excess of the base and the second alkylating agent (CH_3_I, 7.75 g, 55.0 mmol) was added to the solution. The product (2.52 g, 6.0 mmol) was obtained by removing the solvent in vacuo and washing the oil with hexane (3 × 100 mL).

Yield: 54%.

^1^H-NMR (400 MHz, DMSO-d_6_): *δ* 9.54 (s, 1H, NC*H*N), 5.26 (s, 1H, O*H*), 4.78–4.70 (br, 1H, C*H*OH), 4.16 (s, 3H, NC*H*_3_ imidazolium ring), 3.91–3.90 (br, 1H, NC*H*)*,* 3.75 (s, 3H, NC*H_3_*), 3.28 (s, 3H, NC*H*_3_), 2.10–175 (m, 8H, *Cyclohexyl*).

^13^C-NMR (100 MHz, DMSO-d_6_): *δ* 153.20–150.12 (*C*=O), 139.69 (*backbone carbons*, CH_3_N*C*=C), 138.55 (N*C*HN), 107.70 (*backbone carbons*, C=*C*NCH_2_), 70.16 (*C*HOH), 65.47 (N*C*H), 37.24 (N*C*H_3_ imidazolium ring), 34.41–28.72, (*cyclohexyl carbons*), 24.44 (N*C*H_3_ purine ring), 23.67 (N*C*H_3_ purine ring).

MALDI-ToF(*m*/*z*): 293.16120 Da attributable to the cationic part of imidazolium proligand [C_14_H_21_N_4_O_3_]^+^

Elemental Analysis: calculated C 40.01, H 5.04, I 30.20, N 13.33, O 11.42; experimental C 40.11, H 5.04, I 30.30, N 13.13, O 11.42.

##### Synthesis of P-L3, 9-[(2-Hydroxy)ethyl)]-1,3,7-trimethylxantinium Iodide

In a round-bottom flask, under an inert atmosphere, theophylline (2.00 g, 11.0 mmol) and K_2_CO_3_ (3.04 g, 22.0 mmol) were suspended in DMF (40 mL) and stirred for 2 h. Afterward, 2-iodoethanol (2.26 g, 13.2 mmol) was added to reaction mixture, which was stirred for 24 h at 120 °C. The reaction mixture was filtered to remove the excess of the base, iodomethane (7.75 g, 55.0 mmol) was added, and the resulting solution was stirred for another 6 h. The caffeine-derived salt (3.13 g, 8.50 mmol) was obtained by removing the solvent in vacuo and washing the product, a crude oil, with hexane (3 × 100 mL).

Yield: 77%.

^1^H-NMR (400 MHz, DMSO-d_6_): *δ* 9.37 (s, 1H, NC*H*N), 5.16 (m, 1H, O*H*), 4.52–4.50 (m, 2H, C*H_2_*OH), 4.19 (s, 3H, NC*H_3_* imidazolium ring), 3.75 (m, 5H, NC*H*_2_*,* N*C*H_3_), 3.27 (s, 3H, NC*H_3_*).

^13^C-NMR (75 MHz, DMSO-d_6_): δ 153.23 and 150.12 (*C*=O), 139.63 (*backbone carbon*, CH_3_N*C*=C and N*C*HN), 107.03 (*backbone carbon*, *C*=*C*N), 58.35 (*C*H_2_OH), 51.39 (N*C*H_2_), 36.91 (N*C*H_3_ imidazolium *ring*), 31.40 (N*C*H_3_ purine ring), 28.49 (N*C*H_3_ purine ring).

MALDI-ToF (*m*/*z*): 239.11513 Da attributable to cationic part of imidazolium proligand [C_10_H_15_N_4_O_3_]^+^

Elemental Analysis: calculated C 32.80, H 4.13, I 34.66, N 15.30, O 13.11; experimental C 32.50, H 4.15, I 34.86, N 15.25, O 13.24.

#### 3.1.2. General Procedure for the Synthesis of Caffeine-Derived N-Heterocyclic Carbene Silver Acetate Complexes (AgL1OAc, AgL2OAc, and AgL3OAc)

Caffeine-derived silver acetate complexes were produced following the synthetic strategies published in the literature [20,33,34,82,83]. Xanthinium salt (1.0 eq) and silver acetate (2.0 eq) were stirred in dry acetonitrile (30 mL) for eight hours, with the exclusion of light. Then, the mixture was filtered to remove the AgI by-product, and the solvent was removed under reduced pressure to lead to the NHC-Ag complex.

##### Synthesis of AgL1OAc, 1,3,7-Trimethylxanthin-9-[(2-hydroxy-2-phenyl)ethyl-8-ylidene]Ag(I) Acetate

In a round-bottom flask, under an inert atmosphere, imidazolium salt P-L1 (0.44 g, 1.0 mmol) and silver acetate (0.33 g, 2.0 mmol) were dissolved in dry acetonitrile (30 mL) and stirred at room temperature for 8 h with the exclusion of light. The mixture was filtered, and the solvent was removed in vacuum. The silver acetate complex (0.20 g, 0.4 mmol) was obtained as a white powder.

Yield: 40%.

^1^H-NMR (400 MHz, DMSO-d_6_): *δ* 7.41–7.29 (m, 5H, *Ph-group*), 5.62 (s, 1H, O*H*), 4.93–4.89 (m, 1H, C*H*OH), 4.60–4.32 (m, 2H, NC*H*_2_), 4.19 (s, 3H, NC*H*_3_ imidazole carbene), 3.75 (s, 3H, NC*H*_3_), 3.27 (s, 3H, NC*H*_3_), 1.77 (s, 3H, O=CC*H*_3_).

^13^C-NMR (100 MHz, DMSO-d_6_): *δ* 186.36 (N*C*N), 174.51 (CH_3_*C*=O), 153.33–150.39 (*C*=O *purine ring*), 142.45 *(ipso aromatic carbon*, Ph-ring), 140.63 (*backbone carbon*, CH_3_N*C*=C), 128.26, 128.11, 125.77 (*aromatic carbons*, Ph ring), 108.82 (*backbone carbon*, C=*C*NCH_2_), 72.10 (*C*HOH), 57.15 (N*C*H_2_), 37.37 (N*C*H_3_), 31.48 and 28.36 (N*C*H_3_ *purine ring*), 22.96 (O=C*C*H_3_).

MALDI-ToF (*m*/*z*): 705.44220 Da attributable to silver bis-carbenic structure [C_30_H_30_AgN_8_O_6_]^+^.

Elemental Analysis: calculated C 44.92, H 4.40, Ag 22.41, N 11.64, O 16.62; experimental C 44.55, H 4.65, Ag 22.66, N 11.84, O 16.75.

##### Synthesis of AgL2OAc, 1,3,7-Trimethylxanthin-9-[cyclohexan-2-ol-8-ylidene]Ag(I) Acetate

Under an inert atmosphere, in a Schlenk round flask, the caffeine-derived salt P-L2 (0.42 g, 1.00 mmol) and AgOAc (0.33 g, 2.0 mmol) were dissolved in dry CH_3_CN (30 mL) and stirred for 8 h. The mixture was filtered to remove the silver iodide by-product and the silver acetate NHC complex was obtained by removing the solvent under reduced pressure. The complex was recovered as an off-white powder (0.19 g, 0.4 mmol).

Yield: 40%.

^1^H-NMR (400 MHz, DMSO-d_6_): *δ* 5.36 (m, 1H, O*H*), 4.63–4.55 (m, 1H, C*H*OH), 4.36 (s, 3H, NC*H*_3_ imidazole carbene), 3.86 (o, 4H, NC*H*, N*CH*_3_), 3.25 (s, 3H, NC*H*_3_), 2.10–1.75 (m, 11H, *Cyclohexyl* + O=CC*H*_3_).

^13^C-NMR (100 MHz, DMSO-d_6_): *δ* 183.85 (N*C*N), 175.40 (O=*C*CH_3_), 153.23 and 150.22 (*C*=O *purine ring*), 139.67 (*backbone carbon* CH_3_N*C*=C), 109.57 (*backbone carbon* C=*C*NCH_2_), 72.33 (O*C*H), 62.90 (N*C*H), 37.80 (N*C*H_3_), 33.40 and 28.49 (N*C*H_3_ *purine ring*), 34.02–23.55 (*Cyclohexyl group*), 22.82 (O=C*C*H_3_).

MALDI-ToF (*m*/*z*): 673.18138 Da attributable to silver bis-carbenic structure [C_28_H_39_AgN_8_O_6_]^+^ (Ag(NHC)_2_-OH).

Elemental Analysis: calculated C 41.85, H 5.05, Ag 23.49, N 12.20, O 17.42; experimental C 41.80, H 5.10, Ag 23.45, N 12.24, O 17.85.

##### Synthesis of AgL3OAc, 1,3,7-Trimethylxanthin-9-[(2-hydroxy)ethyl)-8-ylidene]Ag(I) Acetate

In a Schlenk round-bottom flask, under a nitrogen atmosphere, P-L3 (0.37 g, 1.00 mmol) and silver acetate (0.33 g, 2.0 mmol) were dissolved in dry acetonitrile (30 mL). The mixture was stirred for eight hours at room temperature with the exclusion of light. Then, it was filtered to remove the silver iodide by-product. The complex (0.22 g, 0.55 mmol) was obtained after removing the solvent in vacuo.

Yield: 55%.

^1^H-NMR (400 MHz, DMSO-d_6_): *δ* 4.73–4.60 (m, 2H, C*H_2_*OH), 4.49 (s, 3H, NC*H_3_* imidazole carbene), 3.96 (o, 5H, NC*H*_2_*,* NC*H*_3_), 3.28 (s, 3H, NC*H*_3_), 1.87 (s, 3H, O=CC*H*_3_).

^13^C-NMR (400 MHz, DMSO-d_6_): *δ* 186.58 (N*C*N); 175.90 (O=*C*CH_3_); 153.03 and 150.42 (*C*=O *purine ring*); 140.72 (*backbone carbon* CH_3_N*C*=C); 108.23 (*backbone carbon*, C=*C*NCH_2_) 60.12 (*C*H_2_OH), 52.99 (N*C*H_2_), 37.69 (N*C*H_3_, imidazole carbene), 31.40 and 28.49 (N*C*H_3_ *purine ring*), 22.82 (O=C*C*H_3_).

MALDI-ToF (*m*/*z*): 583.12378 Da attributable to silver bis-carbenic structure [C_20_H_28_AgN_8_O_6_]^+^.

Elemental Analysis: calculated C 35.57, H 4.23, Ag 26.62, N 13.83, O 19.74; experimental C 35.54, H 4.26, Ag 26.55, N 13.76, O 19.83.

#### 3.1.3. General Procedure for the Synthesis of Caffeine-Based N-Heterocyclic Carbene Gold Iodide Complexes (AuL1, AuL2, AuL3)

Caffeine-derived gold NHC compounds have been synthesized following the procedure developed by Nolan and Gimeno [40,41]. Under a nitrogen atmosphere, the imidazolium salt (1.0 eq), Au(SMe_2_)Cl (1.0 eq), and K_2_CO_3_ (10 eq) were suspended in 25 mL of dry acetonitrile or acetone. The reaction mixture was stirred for 24 h at room temperature. After that, the mixture was filtered, and the solvent was reduced to 5.0 mL. The gold complexes were obtained by precipitation by adding diethyl ether (20 mL).

##### Synthesis of AuL1, 1,3,7-Trimethylxanthin-9-[(2-hydroxy-2-phenyl)ethyl-8-ylidene]Au(I) Iodide

P-L1 (0.22 g, 0.50 mmol) and Au(SMe_2_)Cl (0.15 g, 0.5 mmol) were dissolved in acetone (25 mL). The reaction mixture was stirred for 3 h. After, K_2_CO_3_ (0.67 g, 5.0 mmol) was added, and the mixture was stirred for another 24 h. Subsequently, it was filtered, and the solvent was reduced to ca. 5 mL. The addition of 20 mL of diethyl ether caused the precipitation of the gold complex (0.20 g, 0.3 mmol).

Yield: 60%.

^1^H-NMR (400 MHz, DMSO-d_6_): *δ* 7.41–7.29 (m, 5H, *Ph-group*), 5.62 (s, 1H, O*H*), 5.13 (m, 1H, C*H*OH), 4.60–4.32 (m, 2H, NC*H_2_*), 4.19 (s, 3H, NC*H_3_* imidazole carbene), 3.75 (s, 3H, NC*H*_3_), 3.27 (s, 3H, NC*H*_3_).

^13^C-NMR (400 MHz, DMSO-d_6_): *δ* 184.00 (N*C*N), 153.53 and 150.49 (*C*=O purine ring), 141.75 (ipso aromatic carbon, *Ph-ring*), 139.83 (backbone carbon, C=*C*NCH_2_), 128.36, 127.61, 125.87 (aromatic carbons, *Ph ring*), 107.62 (backbone carbon, CH_3_N*C*=C), 71.60 (*C*HOH), 56.35 (N*C*H_2_), 38.21 (N*C*H_3_ imidazole carbene), 31.68 and 28.36 (NC*H*_3_ purine ring).

MALDI-MS (*m*/*z*): 825.27616 Da attributable to a bis-carbenic gold complex structure [C_32_H_26_AuN_8_O_6_]^+^.

Elemental Analysis: calculated C 30.11, H 2.84, N 8.78, I 19.88; experimental C 30.22, H 2.90, N 8.90, I 19.00.

##### Synthesis of AuL2, 1,3,7-Trimethylxanthin-9-[cyclohexan-2-ol-8-ylidene]Au(I) Iodide

In a round-bottom flask and under an inert atmosphere, the imidazolium salt (0.21 g, 0.5 mmol) and dimethylsulfide gold(I)chloride (0.15 g, 0.5 mmol) were dissolved in dry acetonitrile (25 mL) and stirred for 3 h. After that, the solution of K_2_CO_3_ (0.67 g, 5.0 mmol) was added and the mixture was stirred for 24 h. The reaction mixture was filtered, and the solvent was reduced to ca. 5 mL. The addition of 20 mL of diethyl ether caused the precipitation of the gold complex (0.20 g, 0.3 mmol).

Yield: 65%.

^1^H-NMR (400 MHz, DMSO-d_6_): *δ* 5.56 (s, 1H, O*H*), 5.17 (b, 1H, C*H*OH), 4.26 (s, 3H, NC*H*_3_ imidazole carbene), 4.08 (b, 1H, NC*H*), 3.86 (s, 3H, NC*H*_3_), 3.25 (s, 3H, NC*H*_3_), 2.20–1.75 (m, 11H, *Cyclohexyl*).

^13^C-NMR (100 MHz, DMSO-d_6_): δ 175.65 (N*C*N), 153.33 and 149.22 (*C*=O *purine ring*), 139.67 (*backbone carbon* CH_3_N*C*=C), 108.77 (*backbone carbon* C=*C*NCH), 72.33 (O*C*H), 62.90 (N*C*H), 36.80 (N*C*H_3_ imidazole carbene), 33.40 and 32.65 (N*C*H_3_ *purine ring*), 28.49, 25.42, 25.37, 23.55 (*Cyclohexyl group*).

MALDI-ToF (*m*/*z*): 781.27614 Da attributable to a bis-carbenic gold complex structure [C_28_H_40_AuN_8_O_6_]^+^.

Elemental Analysis: calculated C 27.29, H 3.27, N 9.09, I 20.59; experimental C 27.30, H 3.26, N 9.90, I 20.60.

##### Synthesis of AuL3, 1,3,7-Trimethylxanthin-9-[(2-hydroxy)ethyl)-8-ylidene]Au(I) Iodide

P-L3 (0.19 g, 0.5 mmol) and gold(I) dimethylsulfide chloride (0.15 g, 0.5 mmol) were dissolved in acetone (25 mL) in a Schlenk round-bottom flask. The mixture was stirred for 3 h and potassium carbonate (0.67 g, 5.0 mmol) was added. The reaction mixture was stirred overnight. After, the reaction mixture was filtered, and the solvent was reduced to a ca. 5.0 mL. The gold complex (0.14 g, 0.3 mmol) was obtained by precipitation with diethyl ether (30 mL).

Yield: 55%.

^1^H-NMR (400 MHz, DMSO-d_6_): *δ* 4.83–4.76 (m, 2H, C*H*_2_OH), 4.49 (s, 3H, NC*H*_3_ imidazole carbene), 3.86–3.73 (o, 5H, NC*H*_2_*,* NC*H*_3_), 3.26 (s, 3H, NC*H*_3_).

^13^C-NMR (100 MHz, DMSO-d_6_): *δ* 184.68 (N*C*N); 153.43 and 150.32 (*C*=O *purine ring*); 139.72 (*backbone carbon* CH_3_N*C*=C), 107.23 (*backbone carbon*, C=*C*NCH_2_), 60.22 (*C*H_2_OH), 52.99 (N*C*H_2_), 39.45 (N*C*H_3_), 31.51, 28.24 (N*C*H_3_ imidazole carbene), 31.40 and 28.49 (N*C*H_3_ *purine ring*).

MALDI-ToF (*m*/*z*): 673.18217 Da attributable to a bis-carbenic gold complex structure [C_20_H_28_AuN_8_O_6_]^+^.

Elemental Analysis: calculated C 21.37, H 2.51, N 9.97, I 22.58; experimental C 21.30, H 2.56, N 10.00, I 22.60.

#### 3.1.4. Synthesis of AuL1OAc, 1,3,7-Trimethylxanthin-9-[(2-hydroxy-2-phenyl)ethyl-8-ylidene]Au(I) Acetate

Gold(I) acetate complex was prepared following the procedure published in the literature [12,42] by a metathesis reaction between NHC gold(I) iodide complex (AuL1, 1.0 eq) and silver acetate(1.0 eq). AuL1 (0.063 g, 0.1 mmol) was dissolved in 10 mL of dry dichloromethane. The solution, after the addition of AgOAc (0.020 g, 0.12 mmol), was stirred for 3 h. The suspension was filtered on a pad of Celite to remove the AgI by-product. The evaporation of the solvent led to the gold(I) acetate caffeine-derived complex (0.030 g, 0.05 mmol).

Yield: 50%.

^1^H-NMR (400 MHz, DMSO-d_6_): *δ* 7.39–7.28 (m, 5H, *Ph-group*), 5.62 (s, 1H, O*H*), 5.13 (m, 1H, C*H*OH), 4.60–4.32 (m, 2H, NC*H*_2_), 4.19 (s, 3H, NC*H*_3_ imidazole carbene), 3.75 (s, 3H, NC*H*_3_), 3.27 (s, 3H, NC*H*_3_), 1.79 (s, 3H, O=CC*H*_3_).

^13^C-NMR (100 MHz, DMSO-d_6_): *δ* 173.90 (O=*C*CH_3_), 168.38 (N*C*N), 153.33 and 149.22 (*C*=O purine ring), 142.17 (*backbone carbon*, CH_3_N*C*=C) 140.07 *(ipso aromatic carbon*, Ph-ring), 131.97–125.59 (*aromatic carbons*, Ph ring), (*backbone carbon*, C=*C*NCH_2_) 72.16 (O*C*H), 56.02 (N*C*H_2_), 39.42 (N*C*H_3_ imidazole carbene), 37.37 and 28.36 (N*C*H_3_ purine ring), 22.81 (O=C*C*H_3_).

MALDI-ToF (*m*/*z*): 825.27616 attributable to a bis-carbenic gold complex structure [C_32_H_26_AuN_8_O_6_]^+^.

Elemental Analysis: calculated C 37.91, H 3.71, N 9.82; experimental C 37.90, H 3.80, N 9.83.

Hydrolysis tests

An amount of 5 mg of each compound was dissolved in anhydrous DMSO-d_6_, and an ^1^H-NMR spectrum was recorded. Subsequently, 10% *v*/*v* of D_2_O was added to the sample, and proton spectra were recorded in the span of 72 h, evidencing no changes whatsoever in the signals of the species present in the solution.

#### 3.1.5. General Procedure for A^3^-Coupling (Aldehyde, Amine, and Alkyne) Reaction Promoted by M-NHC (M=Ag, Au) Complexes

In a 10 mL Schlenk tube, aldehyde (1.0 mmol), piperidine (1.2 mmol), phenylacetylene (1.5 mmol), 2-bromomesitylene, as internal standard, and the catalyst (3% mol) were introduced and stirred for 6 h at 80 °C. The reaction mixture was cooled at room temperature. The catalytic activities were calculated by ^1^H-NMR, integrating the signal at 6.89 (2H of 2-bromomesitylene) and proton(s) in α to the nitrogen atom of propargylic amine, i.e., at *δ* 3.43, 2H for *N*-(3-phenyl-2-propynyl) piperidine PAA1; at *δ* 3.11, 1H for 1-(1-cyclohexyl-3-phenyl-2-propynyl) piperidine PAA2; and at *δ* 4.79, 1H for *N*-(1,3-diphenyl-2-propynyl) piperidine PAA3, respectively. The NMR characterization of coupling products is reported in Ref. [43].

#### 3.1.6. General Procedure for Hydroamination reaction between Phenylacetylene and Aniline Promoted by AuNHC Complexes

In a 10 mL Schlenk tube, under an inert atmosphere in dry CH_3_CN (1.0 mL), the aniline (1.0 mmol), phenylacetylene (1.5 mmol), AuNHC (1% mol), and AgSbF_6_ (1% mol) were dissolved. The flask was placed in an oil bath at 90 °C and the mixture was stirred for 16 h.

The solvent was removed, and crude oil was added to the dibromoethane (1.0 mmol) as an internal standard. The yields of imines were determined by ^1^H NMR analysis, after the dissolution of the sample in CDCl_3_. The NMR characterization of the imines is reported in the literature in Ref. [52].

### 3.2. Biology

#### 3.2.1. Cell Culture

The cell lines used were purchased from American Type Culture Collection (ATCC, Manassas, VA, USA). Human breast cancer cells MCF-7 and MDA-MB-231, human mammary epithelial cell line MCF-10A, and mouse BALB/3T3 embryonic fibroblasts were cultured as already described by some of us [84]. The murine macrophage cell line RAW 264.7 was maintained in Dulbecco’s Modified Eagle’s Medium (DMEM) high glucose (4.5 g/L) supplemented with 10% Fetal Bovine Serum (FBS), 1% L-glutamine, and 100 units/mL of penicillin/streptomycin. The cell cultures were maintained at the temperature of 37 °C in a humidified atmosphere containing 5% CO_2_.

#### 3.2.2. MTT Assay

The 3-(4,5-dimethylthiazol-2-yl)-2,5-diphenyl-2H-tetrazolium bromide (MTT) assay (Sigma Aldrich (St. Louis, MO, USA)) was carried out to study the in vitro anticancer activities of the complexes, following the previously reported protocol [85]. The cells were incubated for 72 h with different concentrations of the tested compounds (0.1, 1, 10, 25, 50, 100, and 500 µM) and the IC_50_ values were calculated from the percent (%) of control using the software GraphPad Prism 9 (GraphPad Software, La Jolla, CA, USA).

#### 3.2.3. TUNEL Assay

The terminal deoxynucleotidyl transferase dUTP nick end labeling (TUNEL) assay was performed to detect cell apoptosis, following the manufacturer’s protocols (CF™488A TUNEL Assay Apoptosis Detection Kit, Biotium, Hayward, CA, USA), with some modifications, as previously indicated. Nuclei staining was performed using DAPI solution (0.2 µg/mL, Sigma Aldrich, Milan, Italy). A fluorescence microscope (Leica DM 6000) was used for the cell examinations at 20× magnification. LAS-X software 3.5.7.23225 was employed for the acquisition and processing of the images, which were representative of three independent experiments.

#### 3.2.4. Human Topoisomerase I (hTopoI) Relaxation Assay

The human topoisomerase I relaxation assay was performed as reported in [86]. Shortly, the recombinant human topo I (TopoGEN, Port Orange, FL, USA) was incubated with the super-coiled pHOT1, in the presence or not of the tested compounds, at the indicated concentrations. The reaction products were then allowed to separate through the agarose gel electrophoresis, and ethidium bromide-stained DNA was visualized under UV light.

#### 3.2.5. Anti-Inflammatory Activity

The Griess assay was performed using RAW 264.7 murine macrophages to study the anti-inflammatory activity in terms of NO production.

RAW 264.7 cells were plated in the 48 multi-wells and, after 24 h, treated with the tested molecules (25 µM) and the lipopolysaccharide (LPS, Sigma Aldrich, Milan, Italy) at a final concentration of 1 µg/mL. LPS is able to induce inflammation by stimulating NO production in immune system cells such as macrophages.

After 24 h, the cell medium Griess reagent (Sigma Aldrich, Milan, Italy) was mixed in with a ratio of 1:1. The Griess reagent allowed for the quantification of NO released by macrophages in the treatment medium. The mixture was left for 30 min under agitation and, subsequently, the absorbance at 540 nm was measured using a multiplate reader. The obtained absorbance allowed us to calculate the NO production % compared to the absorbance obtained for the positive control, in which the cells were treated with LPS.

#### 3.2.6. Immunofluorescence

Cells grown on glass coverslips in full medium were serum-deprived for 24 h and then treated with the complexes at a concentration of 25 µM for 24 h. After a PBS wash, cells were fixed using cold methanol (15 min at −20 °C) and incubated overnight at 4 °C with the primary antibodies raised against NF-κB, iNOS, TNF-α, and COX-1 and 2 (Santa Cruz Biotechnology Inc., Santa Cruz, CA, USA) diluted in BSA 2%, as previously described [87]. Alexa Fluor^®^568 conjugate goat-anti-mouse (1:500 dilution), purchased from Thermo Fisher Scientific (Waltham, MA, USA), was used as the secondary antibody. DAPI (Sigma Aldrich, Milan, Italy) 0.2 µg/mL was added for 10 min for nuclei counterstaining. A fluorescence microscope (Leica DM 6000) was used for visualizing images, and LAS-X software 3.5.7.23225 was used for acquisition and processing. The fluorescence quantification was performed using Image J software 1.54d.

#### 3.2.7. Antioxidant Activity

In vitro detection of intracellular ROS on fibroblasts BALB/3T3 using DCFH_2_-DA

The assay to evaluate the in vitro antioxidant activity was carried out using the murine BALB/3T3 fibroblasts that were plated in a white 96 MW and then treated with the complexes at a concentration of 25 µM for 24 h. Menadione (25 µM) was then added for 15 min to induce ROS production. After treatment, the cells were washed with PBS and treated with 25 μM 2′-7′-dichlorofluorescein diacetate (DCFH_2_-DA, Sigma-Aldrich, St. Louis, MO, USA), and then they were incubated for 40 min at 37 °C and 5% CO_2_. The probe DCFH_2_-DA penetrates cells where it is esterified into a non-fluorescent form (DCFH_2_) by endogenous esterases. In the presence of intracellular ROS, non-fluorescent H_2_DCF is oxidized to the fluorescent 2′,7′-dichlorofluorescein (DCF). Then, the cells were washed with PBS and the fluorescence was detected using a multiplate reader (λ_ex_ = 485 nm, λ_em_ = 535 nm).

Based on the obtained fluorescence values, the ROS production inhibition % (*I_ROS_*) vs. menadione was calculated using the following formula:(1)$$I_{ROS}\;vs.\;Menadione\;(\%)=\frac{F_{Men}-F_{Sample}}{F_{Men}}*100$$
where *F_Men_* represents the mean fluorescence obtained after treatment with menadione alone and *F_sample_* represents the mean fluorescence obtained after the co-treatment with the complexes and menadione.

##### 2,2-Diphenyl-1-picrylhydrazyl (DPPH) Assay

The radical scavenging properties of the complexes on the 1,1-diphenyl-2-picrylhydrazil (*DPPH*) radical were evaluated as previously described [86].

All the complexes were used at different concentrations (10, 25, 50, 100, and 200 µg/mL) and the *DPPH* radical scavenging activity, measured at 517 nm using a microplate reader, was expressed as inhibition percentages (%*I_DPPH_*) compared to the initial concentration of *DPPH* (control) according to the following expression:(2)$$I_{DPPH}\;(\%)=\frac{{ABS}_{CTRL}-{ABS}_{sample}}{{ABS}_{CTRL}}*100\;$$
where *ABS_CTRL_* and *ABS_sample_* are the absorbance at 517 nm in the control and in the presence of samples, respectively.

The obtained *I_DPPH_* percentages allowed us to calculate the IC_50_ values using Graph-Pad Prism 9 software (GraphPad Inc., San Diego, CA, USA). Trolox was employed as a positive control, while caffeine and theophylline were reference compounds.

##### 2,20-Azinobis(3-ethylbenzothiazoline-6-sulfonic acid (ABTS) Assay

The radical scavenging properties of the complexes on a 2,20-azino-bis(3-ethylbenzothiazoline-6-sulfonate) radical cation (ABTS^+^) were evaluated as previously described [86].

All the complexes were used at different concentrations (5, 10, 25, 50, and 100 µg/mL) and the ABTS radical scavenging activity, measured at 730 nm using a microplate reader, was expressed as inhibition percentages (%*I_ABTS_*) compared tothe control, according to the following expression:(3)$$I_{ABTS}\;(\%)=\frac{{ABS}_{CTRL}-{ABS}_{sample}}{{ABS}_{CTRL}}*100\;$$
where *ABS_CTRL_* and *ABS_sample_* are the absorbance at 730 nm in the control and in the presence of samples, respectively.

IC_50_ values were calculated from the %*I_ABTS_* using GraphPad Prism 9 software (GraphPad Inc., San Diego, CA, USA). Trolox was employed as a positive control, while caffeine and theophylline were reference compounds.

#### 3.2.8. Minimum Inhibitory Concentration (MIC) and Minimum Bactericidal Concentration (MBC) Determination

The bacterial strains used for MIC and MBC value determinations expressed as µg/mL by the broth dilution method, according to CLSI guidelines [88], were the following:

-4 Gram-negative: *Escherichia coli* (ATCC^®^ 25922^TM^) and *Klebsiella pneumoniae* (ATCC^®^ 13883^TM^);

*Pseudomonas aeruginosa* (ATCC^®^ 15442^TM^) and *Salmonella typhimurium* (ATCC^®^ 14028™);

-3 Gram-positive: *Enterococcus faecalis* (ATCC^®^ 19433^TM^), *Staphylococcus aureus* (ATCC^®^ 23235^TM^), and *Staphylococcus epidermidis* (ATCC^®^ 14990™).

Increasing concentrations of the complexes (50, 70, 100, 150, and 200 µg/mL) were incubated with bacteria plated in 96-well microplates at 37 °C overnight as already reported [86], and then bacterial growth was checked at a wavelength of 600 nm using a Multiskan spectrophotometer (model Multiskan Ex Microplate; Thermo Scientific, Nyon, Switzerland). MIC or MBC values were obtained by comparing cell density with a positive control (bacterial cells grown in LB medium were added with only the vehicle, DMSO). The results were representative of three independent experiments performed in triplicate. Ampicillin, Gentamicin, and Cephalosporin (Sigma Aldrich A9393) were used as the control for strain sensitivity.

#### 3.2.9. Statistical analysis

Data were analyzed for statistical significance (*p* < 0.001) by using one-way ANOVA followed by Dunnett’s multiple comparison test performed by GraphPad Prism 9. Standard deviations are shown.

## 4. Conclusions

Despite the enormous scientific improvements in cancer treatment, cancer-related deaths worldwide are still elevated. The efficacy of clinical anticancer therapies is usually limited by the onset of dramatic side effects and resistance phenomena. Moreover, the presence of inflammation and oxidative stress, closely associated with most of the cancer stages or often induced by the treatment itself, contributes to the creation of an optimal microenvironment for tumorigenesis, development, and metastasis. Thus, the need for novel classes of compounds that can overcome the limitations of the clinically employed treatments and possess multiple biological activities is still critical. At the same time, the development of versatile organometallic compounds, active both biologically and as catalysts for small molecules’ synthesis, is still very topical. In this regard, the use of coinage metal complexes is becoming more and more investigated, given their ease of synthesis, non-inert condition stability, and application versatility. Herein, we reported the design, synthesis, and biological and catalytical properties of a series of new silver(I) and gold(I) complexes, stabilized by a caffeine derivative as NHC ligands. We demonstrated that the individuated leads possess good anticancer, anti-inflammatory, antioxidant, and discrete antibacterial properties, potentially acting contemporaneously on different aspects of tumorigenesis. Finally, these complexes were found to be interesting catalytic agents in *A*^3^-coupling and in hydroamination reactions. When compared to NHC complexes recently reported by us, theophylline’s derivatives resulted as performative in both of the benchmark reactions, and in the case of *A*^3^-coupling, also in neat conditions. Our results highlight some of the potential applications of these complexes and pave the way to the expansion of their structural diversity, aiming to obtain useful and multi-action drug candidates.

## Data Availability

Data is contained within the article and Appendix A.

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
