# Peer review of "Silver and Gold Complexes with NHC-Ligands Derived from Caffeine: Catalytic and Pharmacological Activity"

_ijms, 2024, doi:10.3390/ijms25052599_

Round 1
Reviewer 1 Report
Comments and Suggestions for Authors
The revised manuscript has addressed all issues raised by previous reviewers. Therefore, I would like to recommend this manuscript to publish as its current form in IJMS.
Author Response
Please, see the attachment

Reviewer 2 Report
Comments and Suggestions for Authors
The authors addressed my previous concerns in their resubmitted manuscript. Accordingly, I feel that the manuscript might be suitable for publication in the current form.
Author Response
Please, see the attachment

Reviewer 3 Report
Comments and Suggestions for Authors
The authors have improved their previous submission on the biological activity and catalytic applications of Au and Ag complexes of three caffeine derivatives.
Most of my previous concerns have been addressed, but some minor revisions should be considered before publication:
- Hydrolysis tests meant to determine the long term stability of the metal drugs should be performed in physiological solution (in the presence of chloride ions);
- Line 224-226 should be deleted coming from the template instructions.
Author Response
Please, see the attachment
